# Mental Health Literacy from the Perspective of Multi-Field Experts in the Context of Chinese Culture

**DOI:** 10.3390/ijerph18041387

**Published:** 2021-02-03

**Authors:** Jue Wu, Lin Zhang, Xu Zhu, Guangrong Jiang

**Affiliations:** 1School of Psychology, Central China Normal University, Wuhan 430079, China; wujueppp@126.com (J.W.); lin_zhang@ccnu.edu.cn (L.Z.); grjiang@yeah.net (G.J.); 2Key Laboratory of Adolescent Cyberpsychology and Behavior, Ministry of Education, Wuhan 430079, China; 3Key Laboratory of Human Development and Mental Health of Hubei Province, Wuhan 430079, China

**Keywords:** mental health literacy, Chinese culture, multi-field experts’ perspectives, qualitative study

## Abstract

The study aimed to explore the opinions of multi-field Chinese experts on mental health literacy and further build a comprehensive picture of mental health literacy based on these opinions. Semi-structured interviews were conducted with ten Chinese experts from the fields of psychiatry, clinical psychology, mental health education, and social work. A mixed deductive-inductive thematic analysis was used in the analysis of the qualitative data. The experts noted that mental health literacy applies both to persons with mental illness and the people who help them. The comprehensive view of mental health literacy that emerged from the interviews included knowledge about mental illness, an attitude of acceptance, respectful behavior, and recognition of the importance of getting help. Characteristically, Chinese components of mental health literacy included living in harmony with others and achieving balance in all aspects of life. To the best of our knowledge, this is the first qualitative study of experts’ views of the concept of mental health literacy in the context of Chinese culture. The experts’ responses to the interviews generated a comprehensive view of mental health literacy, including several elements that may be especially salient in Chinese culture. The results have implications for researchers and clinicians.

## 1. Introduction

According to the 2013–2015 National Epidemiological Survey of Mental Illness, the lifetime prevalence of mental illness in China was about 16.6% [1]. Despite this high prevalence, few received mental health services. Phillips et al. estimated at the time of the study that there were 173 million people with various mental illnesses in China, 158 million of whom had never received professional mental health assistance [2]. Mental health literacy (MHL) is one of the most important factors affecting the willingness of people with mental illness to seek help [3].

### 1.1. The Development of the Concept of Mental Health Literacy

MHL is a top-down concept, originating from the views of experts. The concept of MHL was first developed by the Australian psychiatrist Jorm et al. based on the concept of Health Literacy (HL) [4]. HL refers to a set of skills that help people function effectively in the healthcare environment [5]. Jorm et al. defined MHL as the knowledge and beliefs about mental illness that contribute to the recognition, management, and prevention of mental illness [4]. Jorm later elaborated on this definition by describing five components of MHL [3]: (a) knowledge of how to prevent mental illness, (b) recognition of when a psychological disorder is developing, (c) knowledge of help-seeking options and available treatments, (d) knowledge of effective self-help strategies for mild problems, and (e) first aid skills to support others who are suffering from a psychological disorder or are in a mental health crisis. In terms of the definition and components of MHL, we can see that Jorm emphasized on knowledge, especially the knowledge related to the recognition and the treatment of mental illness.

Although Jorm et al.’s original concept of MHL has been regarded as the “gold standard,” some researchers have argued that it had a limitation: it focuses on coping with mental illness but neglects the maintenance of mental health [4,6,7]. In fact, the definition introduced by Jorm et al. is more like “mental illness literacy” rather than “mental health literacy” [4]. Kutcher and colleagues expanded Jorm et al.’s definition of MHL by adding the element of mental health promotion [6]. As a result, the conceptualization of MHL was broadened to include not only the previous focus on coping with mental illness (recognition, treatment, prevention) but also mental health promotion [6,8,9], including self-help and helping others [3,10].

Apart from knowledge, as emphasized by Jorm, many studies have found cognitive factors to be important for MHL [11]. Specifically, people’s beliefs, awareness of stigma, awareness of bias, and help-seeking efficacy can affect people’s ability to help themselves and others to deal with mental illness and maintain mental health [7,12,13,14,15,16]. Therefore, some researchers have argued for expanding the concept of MHL to include not only knowledge but also perceptions (both explicit and implicit), attitudes (including stigma), and the ability to act [6,7,8,17]. Kutcher et al. presented this new, broader definition of MHL [6].

### 1.2. Limitations of the Concept of Mental Health Literacy

MHL is a concept first proposed by psychiatrists [3]. At the early stage, the concept was mainly related to coping with mental illness, which was what psychiatrists were concerned about. Although the concept of MHL has been expanded to a certain degree recently [4,6], this expansion has still been proposed by psychiatrists. It leads to the primary focus in the conceptualization of MHL still being on mental illness, influenced by the field of psychiatry and the Diagnostic and Statistical Manual of Mental Disorders [18]. This phenomenon can be called the fallacy of appealing to authority [19]. Jorm et al. and Kutcher et al. both proposed a concept of MHL that was focused completely or primarily within the framework of psychiatry [4,6]. We admit that psychiatrists are the most suitable experts to evaluate the knowledge component of the concept of MHL, but they may not be familiar with other components, such as beliefs and attitudes (including stigma, help-seeking efficacy), which belong to the realm of ethical judgment usually evaluated by sociologists and ethicists [4,6,17].

### 1.3. Cultural Diversity of Mental Health Literacy

People’s concepts of mental health appear to have both similarities and differences under different cultural backgrounds. A key similarity is a cross-cultural congruence in people’s negative perceptions of people with mental illness. A German study showed that the public perception of people with mental illness, taking schizophrenia as an example, is they are unpredictable, aggressive, dangerous, unreasonable, lacking intelligence, lacking in self-control, and frightening [11]. Another study in China showed that people with mental illness were viewed as unpredictable, bizarre, difficult to communicate with, socially avoidant, needing to be taken care of, and character-deficient [20].

There also appear to be key cultural differences in the concept of mental illness. In Asian cultures, less socially disruptive symptoms of mental illness such as insomnia, mental pain, and low energy are often not treated as mental illness and may not be seen as requiring professional help [21]. Second, there may be cultural differences in people’s understanding of the causes of mental illness. People in Western cultures are more likely than those in non-Western populations to support the biological explanation of mental illness [22]. Third, there may be cultural differences in the perceptions and attitudes of people with mental illness about seeking professional treatment. Compared with White, African Americans have more positive attitudes toward seeking professional mental health services, but are less likely to use them and have less positive attitudes after utilization [23].

### 1.4. Purpose of the Study

Considering the limitations and cultural diversity mentioned above, Jiang et al. proposed a new and comprehensive framework of MHL, which contains two dimensions, namely “coping with mental illness—promoting mental health” and “self—others.” Each dimension contained three aspects: knowledge, attitudes, and behavior [24]. Based on the framework, MHL can be divided into six categories: (1) knowledge about mental illness, (2) attitudes and behavior in responding to others’ mental illness, (3) attitudes and behavior in coping with one’s own mental illness, (4) knowledge about mental health, (5) attitudes and behavior in maintaining and promoting others’ mental health, and (6) attitudes and behavior in maintaining and promoting one’s own mental health.

The purpose of this study was to identify and integrate the perspectives of experts from multiple fields based on the new and comprehensive framework of MHL in the Chinese cultural context. According to the six categories, we conducted semi-structured individual interviews with ten experts who came from the fields of psychiatry, clinical psychology, mental health education, and sociology. The present research is the first qualitative study to explore the components of MHL from the perspective of multi-field experts.

## 2. Materials and Methods

### 2.1. Study Design

This study employed semi-structured interviewing with specific guiding questions along with probing and follow-up questions to ensure an accurate understanding of the experts’ meaning. The interview questions were introduced based on the six components of MHL [24]. Example questions from the semi-structured interview were, “What kind of knowledge do you think a person with high MHL should have when dealing with mental illness?” “What kind of attitudes do you think a person with high MHL should have when dealing with his or her own mental illness? And what should she or he do?” The outline of the interview can be seen in Appendix B, Table A1.

Before the study, we conducted preliminary individual interviews with six mental health practitioners or researchers in order to ensure that the interview outline was logical and comprehensive, and to identify any factors that might hinder the interview’s ability to gather meaningful information. The members of the research team then held several meetings to review the information gathered in these preliminary interviews, determine if changes needed to be made in the wording or organization of the guiding, probing, and follow-up questions, and identify ways to help the interviewees express their views clearly.

### 2.2. Participants

Ten experts (two psychiatrists, four clinical psychologists, two mental health educators, and two experts in social work) were recruited based on these criteria: having done relevant research and published papers in the field of MHL. Four experts were male, and six experts were female, with an average age of 56.0 years (standard deviation (SD) = 12.2). Table 1 summarizes the demographic information of the interviewees. In order to prevent these experts from being identified, we did not show interviewees’ ages in Table 1. The study was conducted in accordance with the Declaration of Helsinki, and the protocol was approved by the Human Research Ethics Committee of Central China Normal University (ccnu-irb-202005-001).

### 2.3. Data Collection

Prior to the formal interview, the participants were informed in advance of the purpose of the study, the content of the interview, and how long the interview would take. Each participant signed an informed consent form. The interviewer and participant then identified a quiet and undisturbed environment in which to do the interview. Interviewees could participate either online or face-to-face, depending on their situations. The formal interviews were conducted using the semi-structured interview outline and lasted for 40–140 min. Interviewees were asked to answer each question as clearly and thoroughly as possible. When the interviewee did not understand the question, the researcher explained it specifically until the interviewee clearly understood the question. The interviews were conducted in Chinese, tape-recorded, and transcribed verbatim. Both interviewers were two graduate students in clinical psychology: one was a master’s degree student, and the other was a doctoral student. They both had acquired basic training in qualitative research and knowledge of mental health and mental illness.

### 2.4. Analysis

A mixed deductive-inductive thematic analysis was utilized for the analysis of the transcripts because the study was theoretically based on the mental health literacy framework proposed by Jiang et al. and its purpose was to determine the specific contents of the MHL framework by investigating the opinions of experts from several fields [24]. The mixed deductive-inductive thematic analysis is appropriate to understand experts’ experiences and perspectives in relation to MHL [25]. The analysis process was conducted rigorously according to the step of mixed deductive-inductive thematic analysis and the criteria for qualitative research (see standards for reporting qualitative research in Appendix A) [26,27].

In the first step, the deductive approach was largely chosen. After reading and re-reading the transcript of the interviews, we developed a preliminary codebook according to Jiang’s s MHL framework [24]. The transcript was coded by using the six broad code categories, i.e., knowledge about mental illness, attitudes and behavior in responding to others’ mental illness, attitudes and behavior in coping with one’s own mental illness, knowledge about mental health, attitudes and behavior in maintaining and promoting others’ mental health, and attitudes and behavior in maintaining and promoting one’s own mental health.

The second step took a largely inductive approach. Specific to the transcript in relation to the six categories, unrestricted open coding was used to generate as many inductive codes as possible. These inductive codes involved minimal interpretation by researchers and were initially summarized in short phrases. For example, a participant’s comments that “You cannot criticize them from the view of moral point, but suggest them to seek help from the professionals, right?” was coded as “No moral judgment”.

Furthermore, these inductive codes were summarized to form subtheme and theme codes. This step involved the researchers’ judgments regarding the relations among the discrete inductive codes. The raters were not bound to the structure of the interviews or questions but instead tried to categorize the inductive codes according to the actual contents of the codes and themes. For example, one interviewee mentioned “No labels” when answering the question regarding the knowledge of MHL. We coded it into the category “attitude”, rather than “knowledge”.

During the analysis process, the initial codebook was refined iteratively according to the findings of inductive codes. The inductive codes were organized iteratively into subthemes and themes within each category according to the codebook. One of the authors analyzed all the transcripts, and the other authors reviewed all the transcripts, themes, subthemes, and inductive codes. To ensure validity, dependability, and establish an inter-coder agreement, we met weekly and resolved disagreements in coding and interpretation through discussion and consensus.

## 3. Results

The experts’ comments about the six components of MHL were integrated to identify several themes. For each category, we first explain the meaning of the category, then describe themes in the category and give some examples.

### 3.1. Category 1: Knowledge about Mental Illness

To help oneself and others to cope with mental illness, the individual should have some basic knowledge, which can be divided into three themes: (1) knowledge about mental illness, (2) knowledge about mental health, and (3) the relationship between mental illness and mental health.

Theme 1: Knowledge about mental illness. To effectively deal with mental illness, most experts agreed that the public should first have a basic understanding of mental illness. The knowledge should cover all aspects of mental illness, including the definition, identification, type, cause, influence, prevention, and treatment of mental illness. For example, expert L mentioned that the causes of mental illness should belong to knowledge about mental illness:

“He/she should know something, too. Where does it (mental illness) come from? What causes it? Of course, ordinary people cannot generalize completely. They probably think about the external environment, or the external pressures, or the people around them. The environment and the atmosphere should relate to some personal characteristics, right? They should have a rough idea of the factors involved.”

Theme 2: Knowledge about mental health. Most experts agreed that to understand what mental illness is, one also needs to understand the corresponding concept of mental health. The knowledge is related to the definition and importance of mental health, and a comprehensive view of mental health. For example, expert Q explained what the comprehensive view of mental health is:

“What kind of concept does a man with good mental health literacy have? I think, first of all, one should have a comprehensive idea of health, which means that health is not just the absence of physical illness, but also the presence of a (good) mental state, adaptive behavior, and good social adjustment.”

Theme 3: Relationship between mental illness and mental health. Whether mental illness and mental health are isolated concepts or served as a continuous spectrum of mental status, there is as yet no agreement among interviewees. For example, expert L stated that the public should have knowledge that it is a continuous spectrum from mental illness to mental health:

“Some people have severe depression, but you want him to go to hospital and get some medicine, he would rather die than go to hospital…if you go to psychiatric department and get psychiatric medications, your life is over. So, it is important to know, everyone has mental health and illness, it should be a continuum, some people are healthier, some people are less healthy”

Expert Z1 said that there is no absolute boundary between mental illness and mental health:

“(Mental) illness and (Mental) health are not absolute concepts, not like the concepts of men and women which belong to absolute classifications, right? It even means that if we make a scale and add some indicators such as body measurement indicators for scoring, and you get 59 that means you have a mental health problem, while I get 61 that means I don’t, right?”

While expert Q suggested that there may an essential difference between mental illness and mental health in the end.

“You know… from mental health to mental illness…it is a process from quantitative change to qualitative change.”

### 3.2. Category 2: Attitudes and Behavior in Dealing with Others’ Mental Illness

The characteristics of an individual who can effectively help others to cope with their mental illness can be divided into two themes: (1) the helper’s traits and attitudes, and (2) the helper’s behaviors and abilities.

Theme 1: The helper’s traits and attitudes. Some experts mentioned that when dealing with people with mental illness, the helper with self-efficacy should be egalitarian, loving, compassionate, empathetic, nondiscriminatory, non-fearful, and respectful. For example, expert L, when mentioning that the helper should treat people with mental illness equally, said:

“…in this aspect (of helping others to cope with mental illness), beyond the theories that we’ve talked about, I think a very important, really the most important thing, is your personality. You should not just see it as a technique, what’s the most important? As Maslow said, it is having a democratic personality. What is a democratic personality? It means that you should respect everyone, right?”

Theme 2: The behaviors and abilities of the helper. To effectively help others to cope with mental illness, a majority of experts mentioned that the public also needed to master some specific behavioral skills, including recognizing others’ mental abnormality, providing emotional support and appropriate realistic help for people with mental illness, and showing self-protection while helping others. For example, when talking about how to provide appropriate realistic help for people with mental illness, expert W mentioned encouraging people with mental illness to seek professional help:

“The helper is not a professional, so he/she doesn’t have to be professional when helping others. The helper encourages the person (with mental illness) to seek help from professional institutions and experts, and I think this is great.”

### 3.3. Category 3: Attitudes and Behavior in Dealing with One’s Own Mental Illness

When an individual has a mental illness, some of their characteristics can effectively help them to cope with it. These characteristics can be divided into two themes: (1) traits and attitudes, and (2) behaviors and abilities.

Theme 1: Traits and attitudes. Most participants thought that self-awareness, self-acceptance, self-responsibility, and an effort towards self-change were the appropriate traits and attitudes that can help patients to cope with their mental illness. For example, when mentioning the self-acceptance of people with mental illness, expert G said:

“Everyone is subject to (genetic and environmental) factors, and there may be various possibilities. We could have heart attacks, psychosis, etc. So, we say that those can possibly happen to other people or me. We should have an open attitude no matter what our bodies become or what our minds become, as these are part of life.”

Besides, expert Z1 mentioned that an attitude may be beneficial to deal with one’s own mental illness:

“…as I mentioned before, it is necessary to avoid self-attribution…”

Theme 2: Behaviors and abilities. To better cope with mental illness, most experts mentioned three kinds of important abilities: self-care, active self-help, and the ability to seek help. For example, expert Z2 talked about engaging in self-help according to one’s situation:

“One can solve some substantive problems in a targeted manner, improve one’s situation, improve one’s ability and expertise, or reduce some pressure. These methods can help the individual out of a dangerous, disadvantaged situation, and they all belong to the area of self-regulation.”

### 3.4. Category 4: Knowledge about Mental Health

To maintain and promote the mental health of themselves and others, people need to master some basic knowledge, which is relatively broad and can be divided into five topics: (1) knowledge about mental health, (2) knowledge about mental subhealth, (3) knowledge about mental illness and psychological problems, (4) the relationships among mental health, mental subhealth, and mental illness, and (5) other professional knowledge.

Theme 1: Knowledge about mental health. A majority of experts believed that the public should be provided knowledge about many aspects of mental health, including the definition and standard of mental health, factors affecting mental health, the impact of mental health, the relationship between mental health and physical health, and the maintenance of mental health. For example, when talking about the definition of mental health, expert L said:

“A person is psychologically healthy, which does not mean that he or she is free from sadness, mental pain, or stress. The same is true for a physically healthy person.”

Theme 2: Knowledge about mental subhealth. In addition to mental health, one expert mentioned mental subhealth, which is the intermediate state between health and illness. The expert believed that the manifestations of mental subhealth include: (1) a pessimistic thinking style, (2) a lot of conflicts in the relationship, (3) physical discomfort, or (4) being easily upset and irritable. For example, when asked about mental subhealth, expert Q said:

“You perceive the world and see things in a more negative and pessimistic way when you are in a state of mental subhealth. We need to see this shift in the person in time. (For example) I was a very energetic personality, and recently became disinterested in everything…There are many manifestations of the subhealth state, such as some physical discomfort, right? Regarding the psychological aspect, there is a state of exhaustion, right?”

Theme 3: Knowledge about mental illness and psychological problems. Mentally healthy people may encounter some psychological problems or even suffer from mental illness in a period of time. In order to maintain and promote mental health, most experts believed that the public needs knowledge about mental illness and psychological problems. Most experts mentioned that knowledge included the definitions, prevalence, controllability, classification, symptoms, causes, impacts, prevention of mental illness, and how to cope with mental illness. For example, expert Z2 mentioned the need for knowledge about the impact of mental illness:

“They should have some knowledge about these (mental illnesses). It is necessary to understand what will happen under extreme pathological conditions.”

Some experts stated that everyone may encounter some psychological problems or suffer from mental illness:

“Don’t think that mental illness is far away, it can happen to any of us!” (Expert Q).

“They (the public) need to know that it is very common for them to have psychological problems.” (Expert F).

Theme 4: Relationships among mental health, mental subhealth, and mental illness. Some experts believed that the public should understand the relationships between mental health, mental subhealth, and mental illness, which can be divided into two types: relationships representing qualitative differences and relationships representing quantitative differences. These two types of differences may be interchangeable:

“… (the public) should know that mental illness results from a process of quantitative change to qualitative change…mental health is relative to mental illness, there is no absolute boundary between mental health and mental illness.” (Expert Q).

Theme 5: Other professional knowledge. In addition to the above knowledge, some experts further mentioned professional knowledge from fields outside their own, including cognitive psychology, general psychology, personality psychology, social psychology, interpersonal relationships, developmental psychology, and psychiatry. For example, expert Z2 argued people should know some knowledge about general psychology and personality psychology:

“For example, cognition, emotion, volition, and personality, etc., are taught in psychology. You should popularize this knowledge so that people can know, for example, why people in the same world have different personalities, different characters.”

### 3.5. Category 5: Attitudes and Behavior in Maintaining and Promoting Others’ Mental Health

The characteristics of people who can effectively help others to maintain and promote mental health can be summarized into two themes: (1) the traits and attitudes of the helper, and (2) the behaviors and abilities of the helper.

Theme 1: Traits and attitudes of the helper. To effectively help others to maintain and promote mental health, most experts said that the public should have attitudes that reflect seeing others as equal and deserving of respect, the belief that one should lead by example, and feelings of altruism, compassion, and tolerance. For example, expert J believed that the public should try to maintain their own mental health and lead by example:

“I think that if one person has a good state of mental health, it will naturally affect others. This may be an unconscious influence, that is to say, he or she exists as a kind of help to others and may impact others simply. If one lives a very positive and optimistic life, he or she will definitely affect others.”

Theme 2: Behaviors and abilities of the helper. To effectively help others maintain and promote mental health, all experts believed that the public also needed to master some specific behavioral skills, such as being kind to others, providing realistic help to others, providing emotional support to others, discovering others’ resources, understanding others, having the ability to help others as a non-professional, recognizing others’ mental illness/psychological problems/psychological crisis, and the ability of self-care. For example, when mentioning discovering others’ resources, expert F argued:

“When someone suffers from a problem, you should listen to him or her first. It’s important to recognize the character strengths and advantages of the people around you, as everyone, no matter how many mental health problems he or she has, always has positive factors that you need to recognize, appreciate and praise.”

### 3.6. Category 6: Maintaining and Promoting One’s Own Mental Health

The characteristics of people who can effectively maintain and promote their mental health can be divided into two themes: (1) traits and attitudes, and (2) behaviors and abilities.

Theme 1: Traits and attitudes. To effectively maintain and promote mental health, all experts mentioned that people needed to view themselves from a developmental perspective, pay attention to their mental health, cherish their mental health, be willing to make efforts for their mental health, take a positive attitude towards themselves, show self-acceptance, and be positive and optimistic. For example, expert Z suggested that people should pay attention to their mental health:

“I mean, we should not wait until there is physical or mental dysfunction to attach importance to mental health.”

Besides, expert L mentioned an attitude regarding the balance between obsession and detachment:

“It is a basic attitude that one should not excessively focus on a certain aspect. Regarding this point, I think the idea from Buddhism is very reasonable, that is what we usually call obsession, which means focusing on one thing and not letting it go, right? It’s a good quality to focus on a goal, isn’t it? But this kind of obsession cannot exceed a certain level, if you cannot achieve this goal, you still want to focus on it, then eventually the loss outweighs the gain, right? Therefore, in this respect, it is sometimes necessary to be both obsessive and detached. Of course, we should have the spirit of persistence, but when you find that you can’t reach your goal, or when circumstances prevent you from achieving it, you should revise your goal. I think that this kind of attitude is good for (mental) health.”

Theme 2: Behaviors and abilities. To better promote and maintain mental health, most experts mentioned several important behaviors and abilities: self-growth, self-awareness, active learning, emotional regulation, maintaining good interpersonal relationships, maintaining a good state of living, the ability to ask for help, and the ability of self-help. For example, expert F stated that constructing a healthy self-image, which was coded as “self-growth”, was important in promoting and maintaining mental health:

“So, what should we do if we want to maintain our mental health? …we need to construct a good and healthy self-image.”

Besides, expert J emphasized a positive and healthy lifestyle for promoting and maintaining mental health:

“…another point is to keep a positive lifestyle…you don’t need to be successful, you don’t have to pursue a career of work, but you are better to keep positive every day, I mean, you feel relatively ok about yourself.”

## 4. Discussion

The present study explored the views of multi-field experts on MHL in the context of Chinese culture. We conducted semi-structural interviews and coded the transcripts based on a mixed deductive-inductive thematic analysis method. To the best of our knowledge, this is the first study to emphasize Chinese culture in the concept of MHL. That is, the present study contributes to our understanding of the cultural factor in MHL. Moreover, it expanded the concept of MHL to a certain extent. The strengths and limitations are introduced below.

### 4.1. Extension of the Concept of Mental Health Literacy

This study explored the six-component model of MHL by interviewing ten experts from multiple professions (psychiatry, clinical psychology, mental health education, and social work) [24]. Through the qualitative analysis of the interview materials, we identified the conceptual framework of MHL proposed by Jiang et al. and expanded the previous concepts of MHL [4,6,24].

In terms of knowledge and the concept of mental illness, in addition to the recognition, treatment, and prevention of mental illness mentioned in previous studies [4], the subthemes regarding causes and effects of mental illness emerged from the interviews in the present study. A previous study showed that the public in China processes information about mental illness according to two core dimensions [28]. One dimension is the severity of the consequence of mental illness, which includes serious damage to the individual and serious damage to society. The other dimension is the personal controllability of the cause of mental illness, which refers to whether the patient is responsible for suffering from mental illness. Both dimensions can influence public attitudes toward mental illness and people with mental illness. In the present study, we found that knowledge about mental illness also included some contents of these two dimensions. For example, some experts argued that the public should develop the concept that mental illness is treatable and controllable and some stated that patients with mental illness should avoid self-attribution. The expansion of these concepts reflects the internal relationships among the components of MHL.

An expert in this study mentioned a special notion in China, i.e., the notion of mental subhealth. Traditional Chinese Medicine (TCM) clinical guidelines for subhealth released by the China Association of Chinese Medicine pointed out that the subhealth status is one that shows declines of vitality, physiological function, and capacity for adaptation, but it is not defined as a clinical or sub-clinical illness [29]. The manifestations of the “mental subhealth” described by the expert in this study are similar to those of the “psychological problem” in the view of the Chinese public. Li has found that the Chinese public has two different core representations of mental health/illness, namely “psychological problems” and “mental illness” [28]. The Chinese public have very different perceptions and attitudes about these two core representations. The public treats “psychological problems” as “no big deal”, with few stigmatizing attitudes, while treats “mental illness” as “catastrophic”, with stigmatizing attitudes. Furthermore, the “mental subhealth” described by the expert in this study reflects a traditional Chinese dialectical thought. Both Chinese experts and the public believe that “misfortune may be a blessing in disguise”. If a bad thing (e.g., mental subhealth) can be easily transformed into a good thing, people do not need to have stigmatizing attitudes towards the bad thing. Therefore, this dialectical thinking probably makes mental subhealth seldom stigmatized.

Concerning attitudes, Kutcher emphasized the need to reduce the stigma on mental illness in order to increase the rate of help-seeking [6]. Many studies have shown that this stigma can lower the willingness of people with mental illness to seek help and lower the willingness of the public to help others cope with mental illness [30,31,32]. The stigma of mental illness includes specific attitudes and emotions toward people with mental illness, such as fear, rejection, anxiety, and aggression [33,34,35]. In this study, the experts not only mentioned attitudes associated with a lack of stigmas, such as empathy, respect, lack of fear, and nondiscrimination, but also described the character traits that contribute to self-help and helping others, such as equality, altruism, optimism, compassion, leading by example, self-acceptance, and self-responsibility. These traits can be viewed as stable attitudes or values that affect people’s relationships with themselves and others (not limited to people with mental illness) [36]. Moreover, in the present study, some experts used the balance theory in traditional Chinese culture to describe what a healthy attitude is. People should be persistent but do not need to be obsessive if they cannot reach the goal.

With regard to the aspect of behavior and abilities needed to seek help and help others, previous researchers paid more attention to professional help-seeking behavior and ability [3,4,17], while the current study added the importance of non-professional behavior and the ability to seek help, to use self-help, and help others, such as providing emotional support and practical help, self-protection, self-awareness, and self-growth. In other words, in addition to advocating professional help, the experts further argued that the public can appropriately help as non-professionals. The experts believed that the public should develop positive attitudes and good habits in their daily lives, strengths which are advantageous for helping themselves and others to cope with mental illness and to maintain and promote mental health. Particularly in the aspect of the maintenance and promotion of mental health, the experts stressed that the public should maintain a healthy lifestyle. Research has indicated that people reporting poor mental health were more likely to report unhealthy lifestyle behaviors, such as smoking and less engagement in physical activity [37].

### 4.2. Comparison of Coping with Mental Illness and Maintaining Mental Health

When comparing coping with mental illness and maintaining mental health, we can find that these two have both similar and different contents. The similarities include (1) the knowledge about mental illness and mental health, (2) attitudes towards oneself and others, such as respect, acceptance, empathy, and understanding others, and (3) behaviors and abilities of self-help and helping others (both professionally and non-professionally). These similarities in the knowledge related to mental illness and mental health suggest that the experts believe that coping with mental illness and maintaining mental health share a focus on knowledge, attitudes, and behaviors.

The different contents of responses regarding coping with mental illness and maintaining mental health are mainly reflected in the knowledge part. The experts believed that coping with mental illness requires knowledge mainly about mental illness, while maintaining mental health requires not only knowledge about mental illness but also professional knowledge from multiple disciplines, such as general psychology, personality psychology, social psychology, interpersonal relationships, developmental psychology, and psychiatry. The experts also expressed the idea that when helping patients cope with mental illness, the individual should pay attention to self-protection and act according to one’s ability. However, this idea did not emerge when talking about helping others maintain and promote mental health.

### 4.3. Differences among Experts’ Opinions

The results of this study indicate that the experts had differences in their understandings of some components of MHL, which were reflected in some of the conflicting themes. For example, the experts all mentioned that the public should understand the relationship between mental illness and mental health. However, they varied in terms of whether they saw mental illness and mental health on a continuum (quantitatively different), as discrete categories (qualitatively different), or some combination There were also differences among the experts regarding the cause of mental illness. Some experts emphasized individual factors and claimed that the individual should be self-responsible; however, other experts focused on the influence of the social environment and argued that the individual should not make self-attributions.

There are likely several reasons for these differences among experts’ views of mental health literacy. First, some experts have a psychopathology perspective, and others have a health psychology perspective on MHL. Second, many questions about mental illness and mental health are still unresolved. For example, is there an essential difference between mental illness and mental health? To cope with mental illness and maintain mental health, is it more accurate (or helpful) to make self-attributions or social attributions of mental illness? Third, because of the limitations of the interview, the experts presented their opinions at a more abstract level, and there may be more agreement among experts from multiple fields when topics are considered at a more concrete level.

### 4.4. Similarities and Differences in the Understanding of Mental Health Literacy between the Experts and the Chinese Public

When we compare the experts’ opinions on MHL in this study with the results of research on the Chinese public’s literacy about coping with mental illness and maintaining mental health [38,39], we find that there are both similarities and differences. The similarities include that both the public and the experts mentioned the topics of recognizing the importance of mental health, mastering some professional knowledge of mental illness, maintaining a healthy lifestyle, accepting one’s own mental illness, being active in self-help and in seeking help, and treating people with mental illness well.

There were two key differences between the public and the experts in their understandings of MHL. First, compared with the public, the knowledge of MHL described by the experts was much broader, including not only professional knowledge of mental illness, but also knowledge of mental health, and the relationship between mental illness and mental health. The experts believed that such knowledge can help individuals deal with their own and other people’s mental illness, as well as maintain and promote their own and other people’s mental health. Second, in contrast to the public, the experts did not emphasize the relationships among morality, values, and mental health. The public believed that a healthy lifestyle needs to be in line with mainstream values and ethical standards; however, in this study, the experts believed that mental illness and psychological problems are not moral problems and people should not make moral judgments about those with mental illness. The moral attribution of mental illness by the public has a long history, which indeed contributes to the feeling of shame and stigma of mental illness [40].

### 4.5. The Cultural Uniqueness of the Concept of Mental Health Literacy

Several components of MHL mentioned by the experts were characteristically Chinese. The experts’ comments also added some perspectives from traditional Chinese culture. For example, the experts mentioned the need to lead by example: people who first maintain their own mental health can then have a positive impact on the mental health of others. In addition to valuing one’s own mental health, one should also “bewail the times and pity the people”, especially for those suffering from mental illness, and the bewailing and pity motivate us to take the initiative to help people with mental illness. The focus on harmonious interpersonal relationships is also characteristic of Chinese culture: one is not an isolated individual, and only in a harmonious interpersonal relationship can one maintain inner psychological harmony and health. Similarly, the goal should be a balance between obsession and rigidness on one hand and detachment and letting it go on the other.

These contents presented by the experts reflect some views in traditional Chinese philosophies, such as emphasizing family, interpersonal relationships, balance, and harmony [41]. In this study, some experts pointed out that we should respect the traditional, effective, non-professional methods of self-help and helping others. This study provides a perspective of the Chinese cultural background for the theoretical research of MHL and extends the concept of MHL to a certain extent, which used to be rooted in the Western cultural background.

### 4.6. Implications

There are at least three implications of the present study. First, this study found that most experts believed that the public should understand the relationship between mental illness and mental health, and some experts used the concept of mental sub-health to explain the intermediate state between mental illness and mental health. Helping the public to realize the connection between mental illness and mental health may reduce their fear of mental illness and their discrimination against mental illness and help them respond with a more positive and rational attitude.

Second, the present study also found that in addition to the knowledge and skills that the previous concepts emphasized, experts argued that mental health literacy should also include some personality characteristics, attitudes, and values, such as sympathy, altruism, and self-acceptance. These are relatively stable for individuals. Therefore, if we add relevant cultivation and promotion of these characteristics, attitudes, and values in basic education and social propaganda, we may effectively improve the public’s mental health literacy level.

Last, most experts in this research believed that MHL should include a healthy lifestyle, which can help individuals maintain and promote mental health. It implies that experts can not only provide basic information or resources for patients with mental illness, but also information or resources about healthy living for the public during public health education, such as organizing mutual aid group activities in the community for people to communicate and share mental health information, paying attention to specific populations under stress and helping them to reduce stress. These can help the public to develop a healthy lifestyle and improve their MHL.

### 4.7. Limitations and Future Directions

The purpose of this study was to explore the understanding of the concept of MHL by multi-field experts in the context of Chinese culture. This study developed a semi-structured interview based on the conceptual framework of MHL proposed by Jiang et al. [24]. The use of a semi-structured interview may have limited the experts’ ability to articulate their understanding of MHL. Another limitation is the small number of participants interviewed for this study. The ten experts participating in this research were from four disciplines. The representativeness of these experts and whether these four disciplines can fully cover all the contents of MHL still need to be further studied and explored.

Future research should address the following issues. Firstly, the consensus among experts from different fields on the concept of mental health literacy can be obtained by using the Delphi method in future studies, based on which evaluation criteria and corresponding measurement tools of MHL can be developed. Secondly, in previous research, the concept of MHL was constructed based on the experts’ theories and standards, which may be largely different from the public’s understanding of MHL. Therefore, it will be helpful to identify differences and similarities in the perspectives of the experts and the public in order to develop a more objective and comprehensive concept, contents, and standards of MHL. Thirdly, by identifying the similarities and differences in the concept and components of MHL in different social and cultural backgrounds, the cultural differences in MHL can be explored.

## 5. Conclusions

This research, conducted in China, is the first qualitative study to explore the components of MHL from the perspective of multi-field experts. The experts’ responses to a semi-structured interview provided a more comprehensive picture of MHL than earlier conceptualizations based only on the psychopathology perspective typical of psychiatry. The results of this qualitative research reveal the specific contents of MHL from the perspective of multi-field experts, many of which reflect the unique characteristics of traditional Chinese culture.

## Figures and Tables

**Table 1 ijerph-18-01387-t001:** Participants’ demographic characteristics (*n* = 10).

Participant	Gender	Field	Area of Expertise	Title
F	Female	Mental Health Education	College students’ mental health and development	Professor
G1	Female	Social Work	Happiness and social support	Lecturer
G2	Female	Clinical Psychology	Healthy mind	Professor
H	Female	Clinical Psychology	The ultimate goal of psychotherapy	Associate Professor
J	Female	Clinical Psychology	College students’ mental health	Professor
L	Male	Mental Health Education	The concept and standard of mental health	Professor
Q	Female	Psychiatry	Survey of mental health knowledge	Professor
W	Male	Clinical Psychology	Crisis intervention	Associate Professor
Z1	Male	Social Work	Psychiatry	Professor
Z2	Male	Psychiatry	Psychosomatic medicine	Professor

## Data Availability

The data presented in this study are available on request to the authors. Some variables are restricted to preserve the anonymity of study participants.

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
