# Peer review of "Mental Health Literacy from the Perspective of Multi-Field Experts in the Context of Chinese Culture"

_ijerph, 2021, doi:10.3390/ijerph18041387_

Round 1

Reviewer 1 Report

This manuscript analyzes the perspectives of several different types of mental health experts on the construct of mental health literacy within the Chinese cultural context.  Overall, the manuscript is clearly written and study appears to be well designed.   While I found the manuscript very interesting, I have some suggestions that, I believe, could improve the manuscript.

1.  On line 134 and again on line 398, the authors write that two of the experts are "sociologists," while in Table 1, they describe them as being in the field of social work.  While there is a great deal of overlap in these fields and a historical connection, the authors should clearly identify whether the respondents are experts in sociology or social work.  The latter typically have greater expertise in the domain of mental health.

2.  In the discussion, the authors discuss (starting on line 420-421) introduce a culturally-specific construct of "mental subhealth" and imply that it is similar to western scientific notions typically describe as psychological distress and distinct from Chinese conceptualizations of mental illness.  This point is a potentially very important culturally distinction that is not carried forward in the discussion nor explicitly elaborated on in the results. How was the notion of mental subhealth evident in these professionals' views?  In the discussion of the stigma of mental illness, it is implied that this stigma may not apply to mental subhealth.  Is this true in the minds of these professionals? the public?  Given the potentially important theoretical contribution of the manuscript, it would strengthen the manuscript if the authors explored this concept further in the analysis and discussion.

Minor editorial suggestions:

3.  On line 123, I would replace "could" with can to be consistent with the present perfect tense used throughout.

4.  The authors could shorten the sentence on lines 508-511 by ending the sentence after "differences" on line 510.  Given the introductory clause, the final phrase seems redundant.

Author Response

We are grateful for taking your time to review this manuscript. We really appreciate all your endorsement, suggestions, and comments. The point-by-point response to the reviewer's comments can be seen in the attachment.

Reviewer 2 Report

Introduction and rationale.

Pertinent literature around Mental Health Literacy is discussed and well critiqued, and cross-cultural differences are considered. The authors illustrate the importance of understanding MHL, stating that it is one of the most important factors affecting the willingness of people with mental illness to seek help. 

What I think was less clear to me, was why the focus on the perspectives of 'experts' on what MHL the Chinese public require or 'should' have in order to be highly literate? I didn't have a clear sense of a rationale for what this hoped to address, and felt that a more pressing question may have been to survey what the mental health literacy of the Chinese public is currently, perhaps focusing in on the MHL of those who have sought help for mental health problems and their families?

The focus on experts opinion on what should be known to be highly literate, seems to take the study in the direction of public health education, and yet there is little mention of how results might be translated into practice or used in this way? 

I'd therefore recommend that a stronger rationale be provided for the particular focus taken by the study. 

Methods. 

The number of participants interviewed for this study is very small, particularly as the sample covers those from different disciplines. This significantly limits the transferability of the findings, a limitation which I think could be more clearly pulled out in the Discussion section.

Choice of Grounded Theory Methods.

I am unclear why the authors chose Grounded Theory (GT) as their qualitative methodology? GT is primarily a qualitative analysis used to generate a theory or model from the data. But the researchers state that this is not their aim. Instead they state that they aim to 'identify the distinct themes within the data' and have therefore just followed GT coding process. 

In execution, there are few aspects of GT methodology demonstrated in the study:

  • The interview questions were guided by pre-existing concepts from the literature, which is not typically recommended in GT methodology.  
  • The interview guide was set at the outset of the interviews rather than developed and evolved as the interviews progressed, as expected in GT methodology. 
  • There was no indication that data collection and analysis happened concurrently as would be expected in GT
  • There was little description of the coding process undertaken, but what is said, i.e. coding descriptively only and then bringing codes together into themes, does not follow the pattern of analysis expected in GT as outlined for example, by Charmaz (2014). 

My conclusion is that there is insufficient evidence that a GT analysis was undertaken, I think it would be far more accurate to describe the methodology as a mixed deductive-inductive thematic analysis. I'd suggest that the authors revisit this section, both to more accurately describe the methodology and to add more detail in terms of the analysis undertaken. A measure of quality in qualitative research, such as Tracy's (2017) Big tent criteria for qualitative research, might also be helpful here, for the authors to reflect upon how they met these standards, and better reflect them in the paper write up.   

Results. 

In the presentation of themes, it is not always acknowledged that the statements are drawn from participants. For example, at the beginning of 3.3 it states: 'Anyone may suffer from mental illness. When an individual has a mental illness, some of their characteristics can effectively help them to cope with it.'  Would it not be more accurate to state something like: 'All/ some/ a majority of participants agreed that anyone may suffer from mental illness... etc'.

I'd therefore recommend that the researchers do more to situate the participants as the source for all the statements in the Results section, and provide some sense of how far themes represent all participants. 

Discussion. 

This section helpfully explores how the current study expands the concept of MHL, how it relates specifically to the Chinese context, and how an expert perspective may differ from the Chinese culture. The authors note that there is some disagreement between professionals in the conceptualisation of mental health/ illness, which is familiar to me in a UK context also. I wondered whether an extension of this study could use a Delphi method to attempt to reach more of a shared position on key aspects? 

One aspect that I felt was lacking in the results section, was any possible applications of this study in a clinical or public setting? What are the implications of this work for public health messaging around mental health/ illness in China? What could these experts understandings of MHL add to how mental health is discussed and understood with those diagnosed with mental illness and with those who support them? Adding in a section around Implications seems particularly and pertinent given the focus and scope of the Journal. 

Author Response

(The authors gave the same response as above.)

Round 2

Reviewer 2 Report

The researchers have addressed my concerns and comments, and I'd support this now going forwards for publication.